# Experimental Study and Development of Design Formula for Estimating the Ultimate Strength of Curved Plates

**Jeong-Hyeon Kim [1], Doo-Hwan Park [1], Seul-Kee Kim [1], Myung-Sung Kim [2] and Jae-Myung Lee [1,2,*]**

[1] Hydrogen Ship Technology Center, Pusan National University, Busan 36241, Korea; honeybee@pusan.ac.kr (J.-H.K.); mabsosa@pusan.ac.kr (D.-H.P.); kfreek@pusan.ac.kr (S.-K.K.)

[2] Department of Naval Architecture and Ocean Engineering, Pusan National University, Busan 46241, Korea; dover@pusan.ac.kr

[*] Correspondence: jaemlee@pusan.ac.kr; Tel.: +82-51-510-2342; Fax: +82-51-512-8836

**Abstract:** The curved plate has been extensively used as a structural member in many industrial fields, especially the shipbuilding industry. The present study investigated the ultimate strength and collapse behavior of the simply supported curved plate under a longitudinal compressive load. To do this, experimental apparatuses for evaluating the buckling collapse test of the curved plates was developed. Then, a series of buckling collapse experiments was carried out by considering the flank angle, slenderness ratio, and aspect ratio of plates. To examine the fundamental buckling and collapse behavior of the curved plate, elastoplastic large deflection analysis was performed using the commercial finite element analysis program. On the basis of both the experimental and FE analysis, the effects of the flank angle, slenderness ratio, and aspect ratio on the characteristics of the buckling and collapse behavior of the curved plates are discussed. Finally, the empirical design formula for predicting the ultimate strength of curved plates was derived. The proposed empirical formula is a good indicator for estimating the behavior of the curved plate.

**Keywords:** buckling experiment; curved plate; ultimate strength; design formula

## 1. Introduction

A cylindrically curved plate is an extensively used main member in many industrial fields such as shipbuilding, offshore, automotive, and aerospace industries. In the ship and offshore industries, the curved plate is the most widely used structural member, normally used in applications such as deck plating with a camber, side shell plating at fore and aft parts, and bilge circle part. Thus, it is necessary to estimate the buckling strength of a curved plate, because it can be subjected to longitudinal compression under hull girder bending and hydrostatic pressure by seawater conditions. In addition, buckling related issues of curved plates have become more important recently because conventional thick structural steel plates have been replaced with the plates that are relatively thin and made of high tensile steel.

In the ship and offshore industries, approval from the classification society is mandatory for shipbuilding. For this purpose, the classification societies have provided design formulae to estimate the buckling strength of curved plates based on the results of commercial finite element analysis. However, there are differences in the design values between the results calculated by the rule given by the classification societies and those obtained from FE analysis. In a previous study, Kim et al. [1] investigated the buckling and ultimate strengths of simply supported curved plates subjected to combined axial compression and lateral pressure by performing a series of elastic and elastoplastic large deflection analyses using commercial finite element program MSC PATRAN/NASTRAN. The numerical buckling analysis results showed that buckling strength and collapse behavior depend on the curvature, initial deflection, slenderness ratio, and aspect ratio. In addition, by comparing the results as per the classification rules and the results of FE analysis, it has

been confirmed that the results of the critical buckling strength show significant differences for thinner plates; as a result, the improved design formulae are suggested.

In general, this difference occurs because of the absence of an appropriate design procedure for estimating the buckling characteristics of a curved plate, as well as experimental investigations to verify the procedure. For this reason, it is necessary to perform research on the development of an appropriate design procedure as well as conduct experimental investigations to validate the results of FE analysis, and to refine the design procedure. Information on the buckling behavior may be useful when the buckling behavior of curved plates with reduced thicknesses due to corrosion is considered. Therefore, the establishment of an experimental procedure for curved plates and the development of a new design formula are tasks in the ship and offshore industries that must be addressed.

In this regard, a few studies that experimentally investigated curved plates have been conducted. Shanmugam reported the results of ultimate strength tests on curved plate girders with circular web openings under transverse compression. It is confirmed that the ultimate load capacity of the girder reduces when the opening and degree of curvature increases in size. In addition, these experimental results are compared with FE analysis using commercial finite element program ABAQUS [2]. Zhang et al. performed experiments on six test models of prestressed concrete curved slabs with unbonded tendons. Through validation with the experimental results, a special computer program for estimating the ultimate strength of target structures was developed [3]. Wang et al. observed residual buckling distortion of a mild steel test specimen, which was fabricated by bead-on-plate welding. The experimental result was compared with the result of FE analysis to verify the effectiveness. It was concluded that the tendon force (longitudinal inherent shrinkage) was the dominant reason for buckling and that the initial disturbance (initial deflection or inherent bending) triggered buckling, whereas the buckling mode was not affected [4].

Despite these efforts, the above-mentioned studies have limitations, in that the estimation of the fundamental buckling behavior of curved plates is difficult because only a few experiments have been performed, and the target plates were highly complex structures. Therefore, in the design of the ship and offshore structures, the understanding of the basic structural response caused by the buckling phenomenon of the main components is inevitably based on the actual experimental database.

This study is a continuation of the authors' previous research, which was conducted to validate and improve the design formulae derived from numerical analysis of curved plates [1]. Thus far, a series of buckling collapse experiments were carried out considering the actual states of the curved plate used in shipbuilding. To investigate the ultimate strength and collapse behavior of a cylindrically curved plate under a compressive load and in the simply supported condition, the main variables considered for the curved plate are the flank angle (θ), thickness (t), and aspect ratio (a/b); a total of 72 plates were tested. In addition, an elastoplastic large-deflection analysis was performed under the same condition as in the experimental preparation, and its results were compared with the experimental results. Finally, new empirical design formulae for predicting the ultimate strength of curved plates were derived and numerical results of previous study were complemented based on the experimental results.

## 2. Experimental Preparations

### 2.1. Test Specimen and Scenario

In the present study, a simple curved plate without stiffeners under longitudinal compression, as shown in Figure 1, is considered for the tests. The dimensions of the cylindrically curved plates are length (a), breadth (b), thickness (t), and flank angle (θ). The breadth (the length of an arc in a cylindrically curved plate) is kept constant at 400 mm throughout the experiment and throughout the finite element analysis.

To investigate the ultimate strength characteristics of the curved plates and obtain an experimental database, three main variables (i.e., flank angle, thickness, and aspect

ratio) are considered. In cylindrically curved plates, the flank angle is the most important parameter. The breadth and flank angle are related by the following relationship:

$$b = R\theta \tag{1}$$

where $R$ is the radius of the curved plate. Three values (2000 mm, 4000 mm, and 8000 mm) extensively used in merchant vessels were selected for the radii. In addition, the aspect ratios in the range of 1.0 to 2.0 and thickness in the range of 6 mm to 8 mm were considered. Based on these main parameters, the experimental scenario is summarized in Table 1. According to this scenario, a series of buckling collapse tests was carried out on a total of 72 plates (two tests for each scenario), consisting of 54 curved plates and 18 flat plates under longitudinal compression. The displacement control mode was used for the buckling experiment. In addition, 1 mm/min loading speed of the universal testing machine (UTM) was considered.

Tensile tests were performed to apply the precise values of material properties. Generally, the chemical composition of the material has a dominant influence on its mechanical properties. The specimens for material tensile tests were extracted from the plate for buckling collapse tests. In the present study, six tensile tests were performed in accordance with American Society for Testing and Materials (ASTM) standard A370. Figure 2a shows the representative tensile testing specimens with three different thicknesses, and Figure 2b shows the preparation of the material tensile tests. Table 2 shows the chemical composition of the selected material, and Table 3 shows the summary of the material tensile tests.

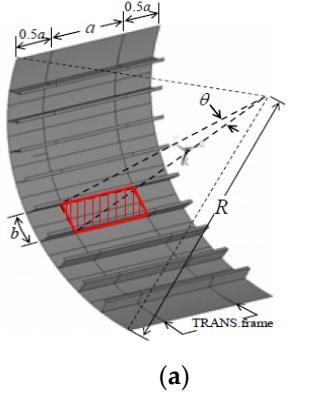

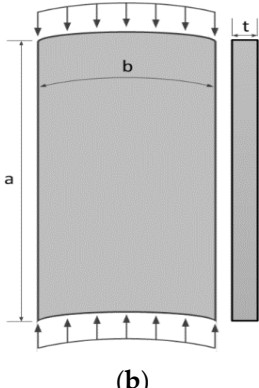

(a)  (b)

**Figure 1.** (**a**) Schematic of a cylindrically curved panel and (**b**) plate.

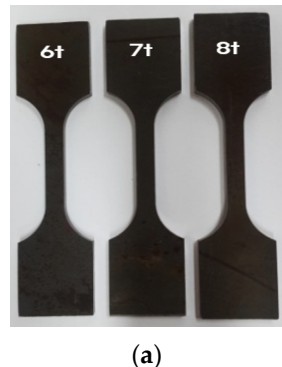

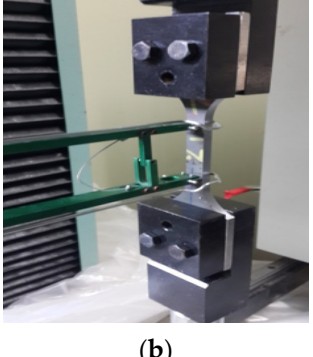

(a)  (b)

**Figure 2.** (**a**) The representative tensile testing specimens with three different thickness and (**b**) image of the test jig.

**Table 1.** Experimental scenario.

| Test Model | a/b (b = 400 mm) | t (mm) | R (mm) |
|---|---|---|---|
| F01 | | | 0 |
| C01 | | | 2000 |
| C02 | 2.0 | 6 | 4000 |
| C03 | | | 8000 |
| F02 | | | 0 |
| C04 | | | 2000 |
| C05 | 2.0 | 7 | 4000 |
| C06 | | | 8000 |
| F03 | | | 0 |
| C07 | | | 2000 |
| C08 | 2.0 | 8 | 4000 |
| C09 | | | 8000 |
| F04 | | | 0 |
| C10 | | | 2000 |
| C11 | 1.5 | 6 | 4000 |
| C12 | | | 8000 |
| F05 | | | 0 |
| C13 | | | 2000 |
| C14 | 1.5 | 7 | 4000 |
| C15 | | | 8000 |
| F06 | | | 0 |
| C16 | | | 2000 |
| C17 | 1.5 | 8 | 4000 |
| C18 | | | 8000 |
| F07 | | | 0 |
| C19 | | | 2000 |
| C20 | 1.0 | 6 | 4000 |
| C21 | | | 8000 |
| F08 | | | 0 |
| C22 | | | 2000 |
| C23 | 1.0 | 7 | 4000 |
| C24 | | | 8000 |
| F09 | | | 0 |
| C25 | | | 2000 |
| C26 | 1.0 | 8 | 4000 |
| C27 | | | 8000 |

**Table 2.** Chemical compositions of the target material.

| Ingredient List | C | Si | Mn | P | S | Ni | Cr |
|---|---|---|---|---|---|---|---|
| Component content (%) | 0.150 | 0.220 | 0.900 | 0.010 | 0.004 | 0.010 | 0.020 |

**Table 3.** Mechanical properties of the curved plate.

| No. | Thickness | Yield Strength | Tensile Strength | Elongation |
|---|---|---|---|---|
| 1 | 6 mm | 364 MPa | 491 MPa | 32% |
| 2 | 6 mm | 345 MPa | 478 MPa | 35% |
| 3 | 7 mm | 332 MPa | 475 MPa | 37% |
| 4 | 7 mm | 342 MPa | 475 MPa | 36% |
| 5 | 8 mm | 333 MPa | 464 MPa | 37% |
| 6 | 8 mm | 343 MPa | 476 MPa | 35% |
| Average | | 343 MPa | 477 MPa | 35% |

## 2.2. Apparatus

Curved plates in steel-plated structures are normally supported by various types of adjacent plates along the edges to provide the rotational and fixed constraints in the out-of-plane direction. In terms of structural design, adoption of the idealized simply supported boundary conditions provides a suitable safety factor compared with the clamped boundary conditions [1]. In the structural design aspect, in order to provide a suitable safety margin, the boundary conditions of the curved plates are considered to be simply supported on all the edges.

To achieve this condition, proper experimental jigs and fixtures were designed based on the reference studies [5–7]. The assembled fixtures with a sample specimen are shown in Figure. 3. A V-shaped groove is made in the fixture and the edge of the curved plate is located in the groove. In order to match the radius of each plate, three pairs of upper and lower jigs with radii matching to those of the specimens were produced. In addition, the loading condition of longitudinal compression was implemented by fixing the side and lower jigs and moving only the upper jig. To prevent the movement of side jigs (Jig I) while the specimen was subjected to a compressive load, steel bands (Jig IV) were installed in the top and middle parts of the jig, as shown in Figure 3b. It was designed to use the side jigs to maintain straightness while the load was applied. The misalignment of straightness was a phenomenon that occurred during or after removal from the jig due to the relaxation of internal stress.

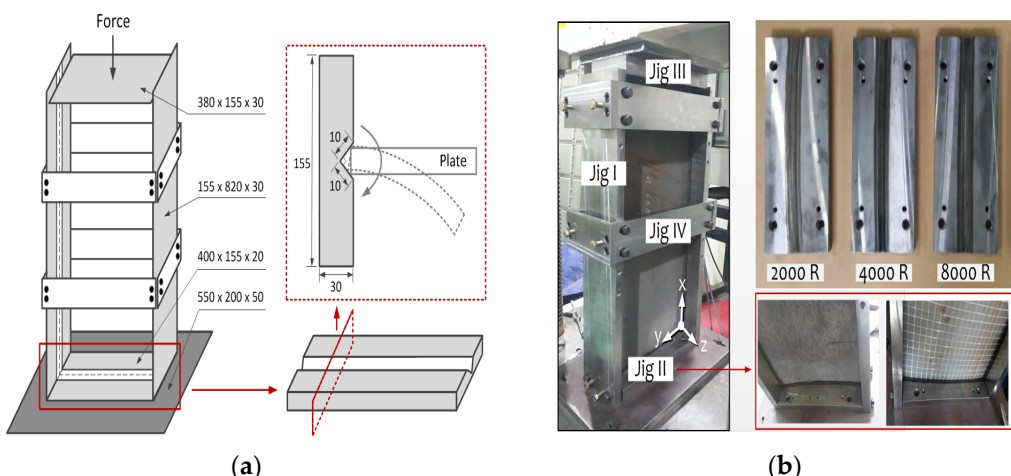

(**a**)                                              (**b**)

**Figure 3.** (**a**) Schematic overview of the experimental jig and (**b**) image of the testing jig.

With these experimental fixtures, a series of buckling collapse experiments was performed using a universal testing machine (UH 1000 kN, SHIMADZU). Figures 4 and 5 show the specimens of curved plates and buckling collapse tests, respectively, after testing.

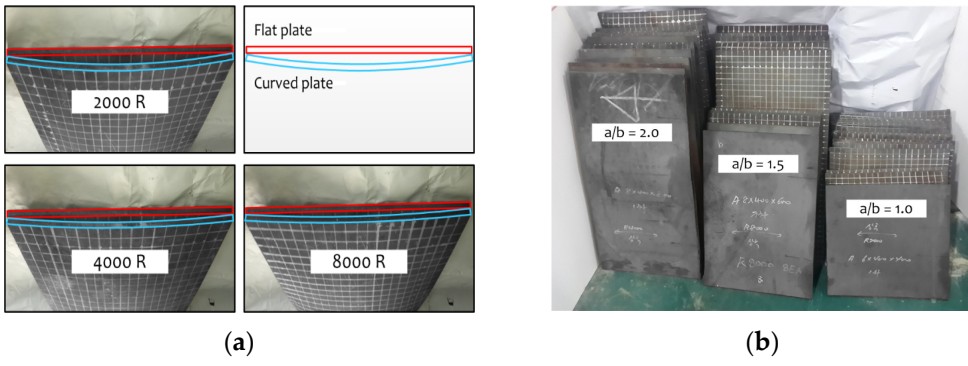

(**a**)                                              (**b**)

**Figure 4.** (**a**) Curved plates specimens and (**b**) test plates after buckling test.

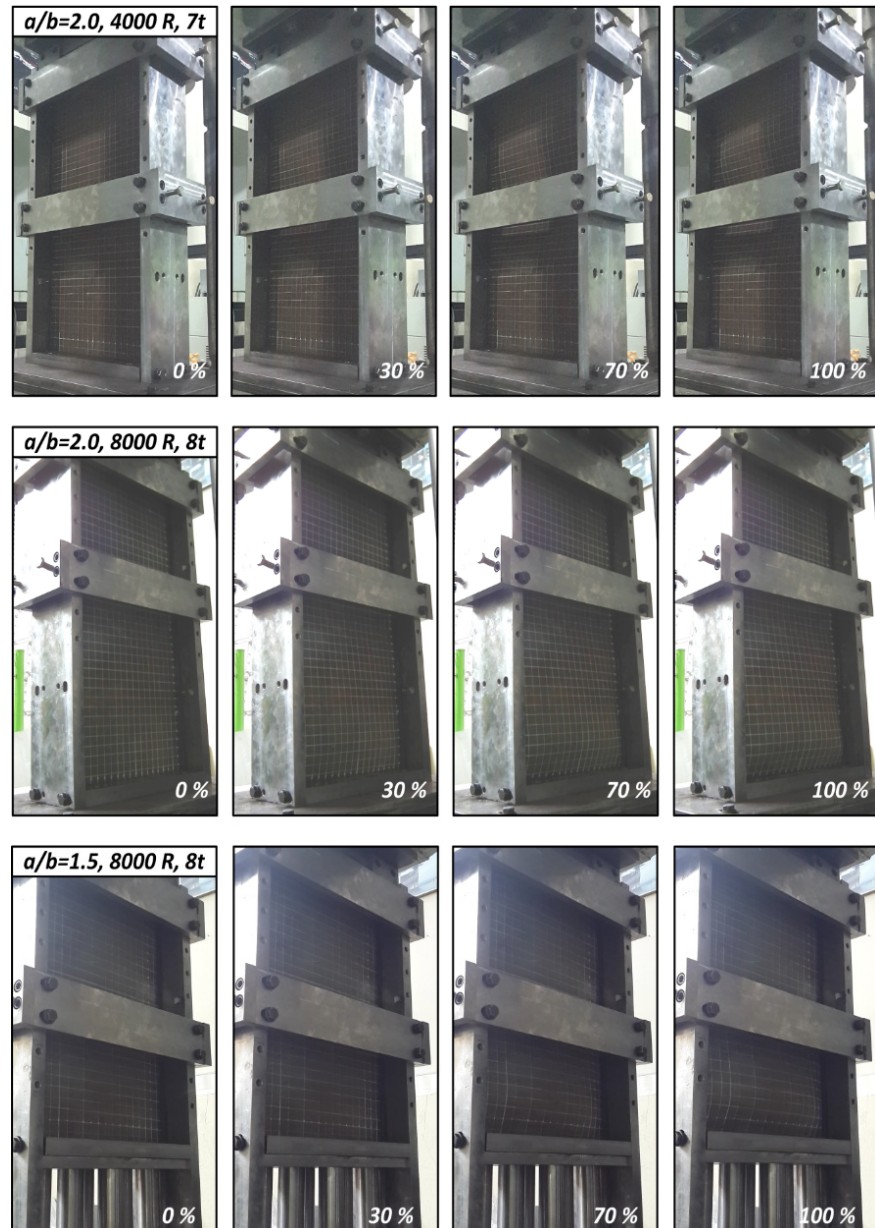

**Figure 5.** Buckling collapse test for curved plates.

### 2.3. Experimental Results

The ultimate strength characteristics, which depend on three main variables of a curved plate (flank angle, slenderness ratio, aspect ratio), were investigated. When the arc length (b) was constant, a small radius means that curved plate bent more, with a large flank angle. The radii selected for the plates, and the equivalent flank angles are as follows: $11.45°$ (R = 2000 mm), $5.72°$ (R = 4000 mm), and $2.86°$ (R = 8000 mm).

Figure 6a shows compressive load versus displacement curve when the thickness is 6 mm and aspect ratio is 2. From this graph, it is confirmed that a curved plate with the highest flank angle has the highest ultimate strength when compared with the curved plates having other flank angles. In addition, the slope of the load-displacement curve after reaching the ultimate strength decreased rapidly when compared to the corresponding curve of a flat plate. This means that once the strength of the curved plate has reached its maximum value, its load-carrying capacity reduces rapidly. This phenomenon is more gradual when the flank angle of the curved plate decreases.

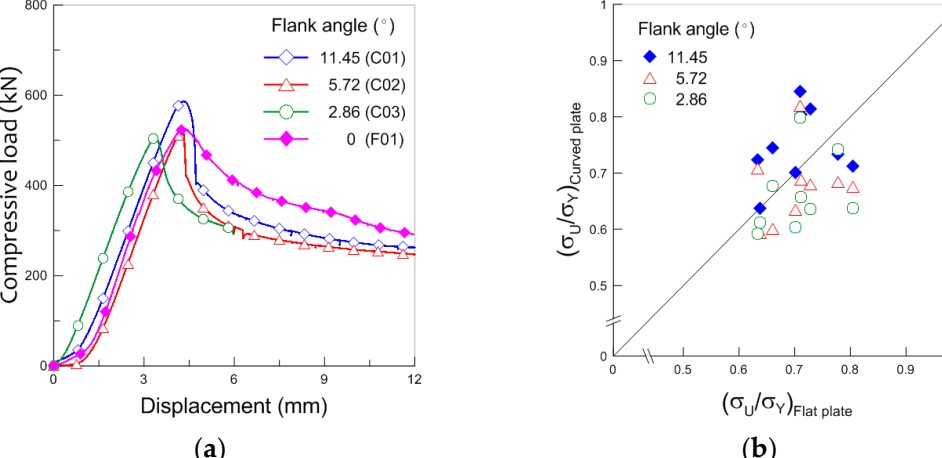

**Figure 6.** (**a**) Compressive load versus displacement curves of curved plates and (**b**) comparison of ultimate strengths of curved and flat plates.

Figure 6b shows the comparison of the results of ultimate strengths with curved and flat plates. In this figure, a clear difference is observed in the ultimate strength distribution of the curved plates near the flank angle of 5°. The ultimate strength of curved plates with a flank angle near or lower than 5° (2.86°, 5.72°) is lower than that of flat plates, while the ultimate strength of curved plates with a flank angle higher than 5° (11.45°) is greater than that of flat plates. In order words, the ultimate strength for a curved plate with a small flank angle (less than 5°) is lower than that of a flat plate under longitudinal compressive loading condition. These results were reconfirmed through another study, which verified that the increase in the ultimate strength is greater when the curvature is higher. Tran et al. [8] reported that the curvature increases the elastic buckling resistance of the curved plates. Thus, the higher the curvature, the higher the effect. In addition, the curvature also changes the shape of bucking: the number of half-waves increases with the curvature.

In addition to the ultimate strength, the buckling mode is distinguished based on flank angle 5°, as shown in Figure 7. When the flank angle is more than 5°, buckling takes place with three longitudinal half-waves. With a decrease in the flank angle, the buckling mode is changed from one longitudinal half-wave to two longitudinal half-waves when the flank angle is zero. These characteristics related to the buckling mode are similar to those of the flat plates (two longitudinal half-waves) as the aspect ratio decreases.

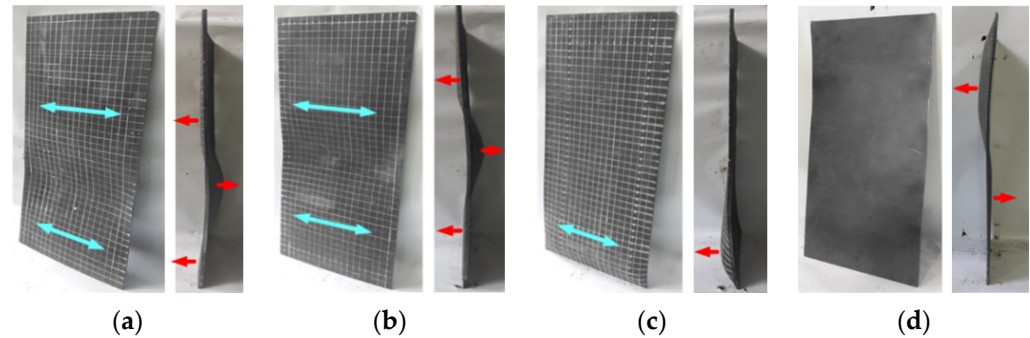

**Figure 7.** Collapse mode of curved plates after testing (a/b = 2.0, t = 6 mm): (**a**) C01, (**b**) C02, (**c**) C03, and (**d**) F01.

Figure 8 shows the effect of flank angle, slenderness ratio, and aspect ratio on the ultimate strength of curved plates based on the results of our experimental investigation. In Figure 8a, the average values of the experimental results for each condition are indicated using a solid line. From the figure, it can be confirmed that an increase in the flank angle causes an increase in the ultimate strength of curved plates. However, in the case of the

slenderness ratio, the ultimate strength displays an unusual trend; the ultimate strength of relatively thin plates (t = 7 mm) is higher than that of relatively thick plates (t = 8 mm). This tendency is more prominent when the a/b ratio is 1.5 or when the flank angle is 11.45° (See Figure 8c).

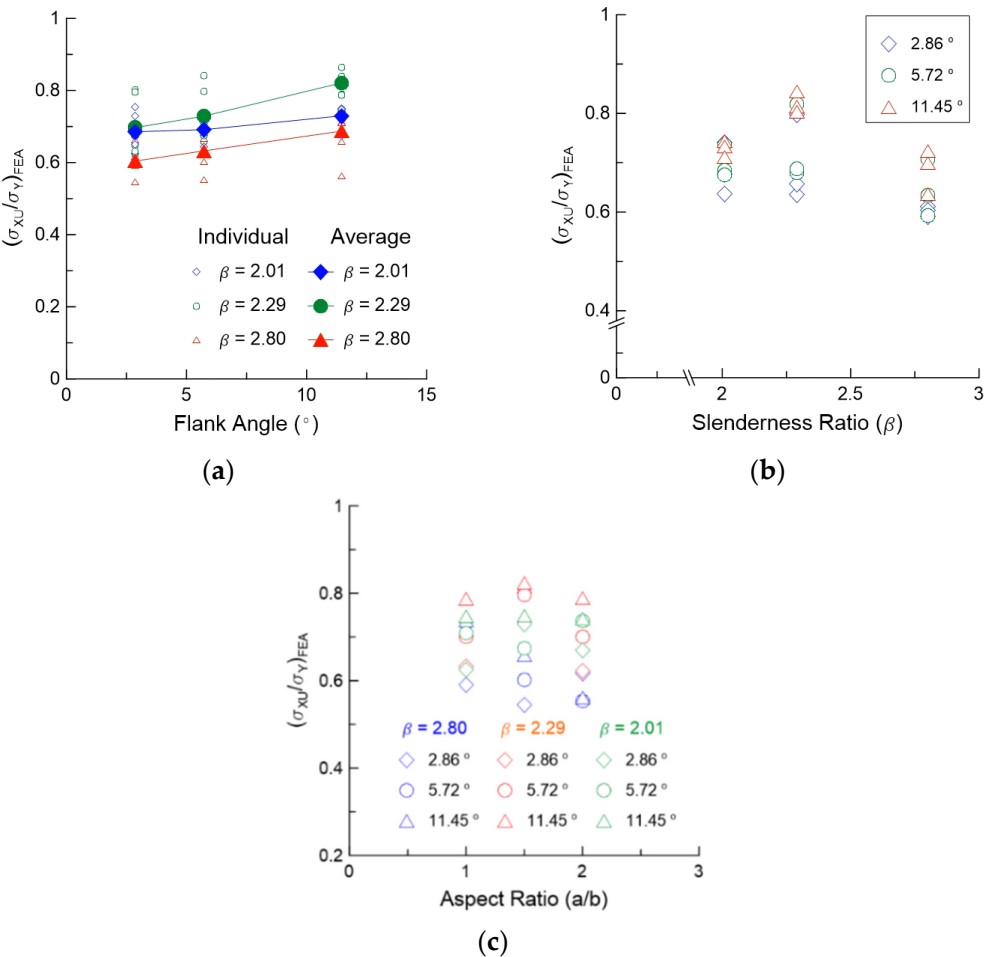

**Figure 8.** Ultimate strength of the curved plates based on the results of experiments: (**a**) effect of flank angle, (**b**) slender-ness ratio, and (**c**) aspect ratio.

In this section, the normal and unusual buckling behaviors of curved plates were identified through the buckling experiments. In the experiments, a unique behavior of curved plate was observed, in that the ultimate strength of the curved plates unexpectedly changed at a certain flank angle (5°), slenderness ratio (2.01) or aspect ratio (1.5). This will be compared with the results of FE analysis, and the reason for this phenomenon is discussed later.

## 3. Design Formula

### 3.1. Investigation for Classification Rule

To explain the fundamental buckling and post-buckling behaviors of curved plates subjected to various loads, the classification societies have suggested equations for the buckling strength of curved plates. Figure 9 shows the comparison of the results as per the classification rules and the results of the FE analysis that was performed using the commercial program ABAQUS for predicting the critical buckling strength of the curved plates. In the case of thicker plates, the critical buckling strengths from the two methods showed good agreement. However, for thinner plates, the results of the critical buckling strength showed significant differences because the curvature reduction factor used in the

classification rule equations do not correctly reflect the buckling phenomenon [9]. For this reason, much research has been conducted using FE analysis to more accurately predict the behavior of curved plates to reflect the curvature effects. FE analysis is performed in order to compare the analysis results with the experimental results and confirm the fundamental buckling behavior of curved plates.

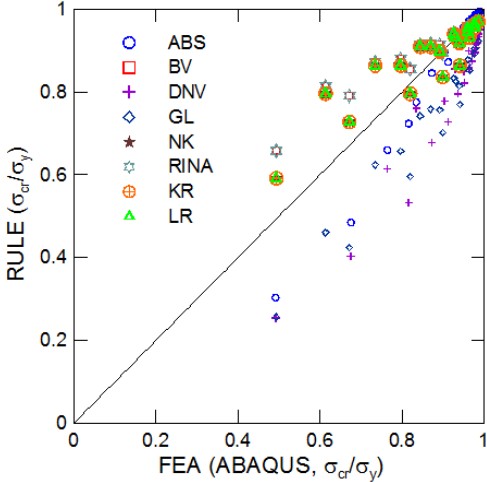

**Figure 9.** Comparison of the results of buckling strength as per classification rules and finite element analysis [1].

### 3.2. FE Analysis

For predicting the nonlinear buckling behavior, the arc-length method was selected as the numerical method in the commercial finite element program (MSC Patran/Nastran 2008). The arc-length method can track complex buckling behaviors such as secondary buckling and snap-through phenomena of thin plates. Thus, it is suitable for predicting the complete load-displacement relationship and deriving a nonlinear static solution. A detailed numerical formula and method can be deduced from previous studies [9,10].

The element type used for the FE analysis of the plates were the 2D surface shell elements Quad4, and the number of elements was 800. The minimum mesh size and the number of elements were determined through a mesh convergence study as given in a reference [1]. In addition, an idealized elastic-perfectly plastic model (yield strength: 343MPa) was selected as the analysis model, and the strain hardening effect was neglected. In terms of material nonlinearity, the plastic potential was evaluated based on the von Mises yield criterion.

The boundary and loading conditions are considered with the experimental conditions, where all the plate edges are simply supported and subjected to longitudinal compression, as shown in Figure 10. Multi-point constraint (MPC) elements (REB2) are used to regularly move the dependent points to the specific points based on an independent point on the basis of the cylindrical coordinate system. In addition, the initial geometric imperfection follows the elastic buckling mode drawn by elastic analysis, and the maximum magnitude of the initial imperfection was assumed to be at the normal level as calculated using Smith's equation ($\omega = 0.05\beta^2 t$) [11].

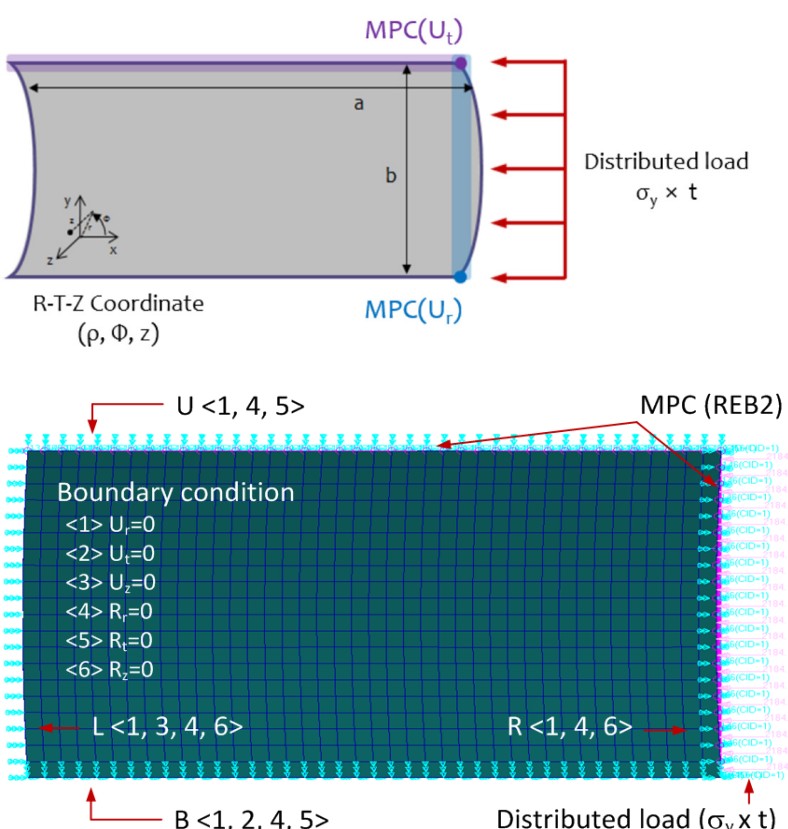

**Figure 10.** Loading and boundary conditions of the curved plate.

## 4. Comparison with Experimental Results

### 4.1. Buckling Mode

Figure 11 shows the buckling shape of the curved plate for various flank angles when the a/b ratio is 2.0 and slenderness ratio is 2.80. When the a/b ratio is 2.0, the buckling mode of a rectangular flat plates has two longitudinal half-waves. As the flank angle increases to below 3°, the central part of curved plates maintains the same half-wave with relatively lower lateral deflection than that of a flat plate. When the flank angle is 5°, the longitudinal half-waves of the curved plate changes from two half-waves to three half-waves. With a further increase in the flank angle, from 5.72° to 11.45°, the buckling mode changes to a single half-wave with bulged components of deflection near the transverse edges. Through a previous study using FE analysis, it is confirmed that the buckling mode of simply supported curved plates under longitudinal compressive loading changes to single half-waves when the flank angle reaches 10° [1]. Thus, as the flank angle increases, the buckling mode sequentially changes to two-three-one longitudinal half-waves, and a single half-wave is maintained when the flank angle is more than 11.45°.

This behavior can be observed in the experimental results in Figure 7. However, there is a difference in that the buckling mode of the test specimen (C01) with a flank angle of 11.45° was not a single longitudinal half-wave. Nonetheless, it is expected that the buckling mode in the experiments sequentially changes to two-(one)-three-one longitudinal half waves.

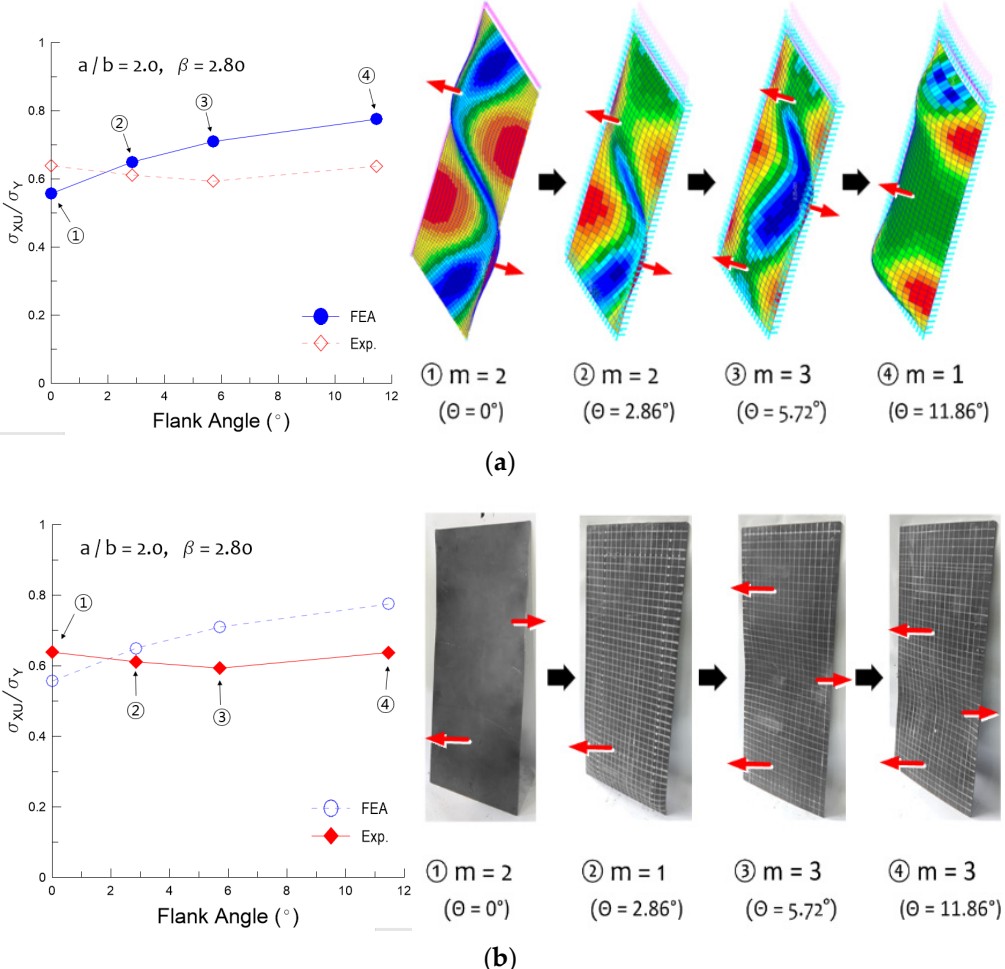

**Figure 11.** The buckling shapes of curved plates depending on the flank angle: (**a**) FE analysis and (**b**) experiment results.

### 4.2. Ultimate Strength

Figure 12 shows the ultimate strength of the curved plates for various flank angles, slenderness ratio, and aspect ratio based on the result of the FE analysis. From these graphs, it can be confirmed that with the increase in flank angle and decrease in slenderness ratio, the ultimate strength of simply supported curved plates increases. However, as outlined above, this trend is different from the experimental results in which the ultimate strength of the curved plates unexpectedly changed at a certain flank angle (5°), slenderness ratio (2.01) or aspect ratio (1.5). This phenomenon can be explained through the previous studies, as given below.

It is well known that the ultimate strength of a curved plate under longitudinal compression loading increases as the flank angle of the curved plate increases. When the compressive load equals the critical buckling stress, the plate starts to buckle. Increasing the axial load further causes an increase in the lateral deflection at the central part of the plate, and the increase in the lateral deflection causes an increase in the membrane stress; this in turn, causes stretching of the surface of plate at the bulged parts near the short edges, as shown in Figure 13 [12]. In addition, as shown in Figure 13c,d, an additional membrane stress is generated in a longitudinal direction at the central part of the curved plate when the flank angle is changed from 5° to 10°. This tensile membrane stress can restrain the lateral deflection of the plates and, as a result, a strength of the curved plate can be significantly improved when that curved plate has a flank angle in excess of 5°.

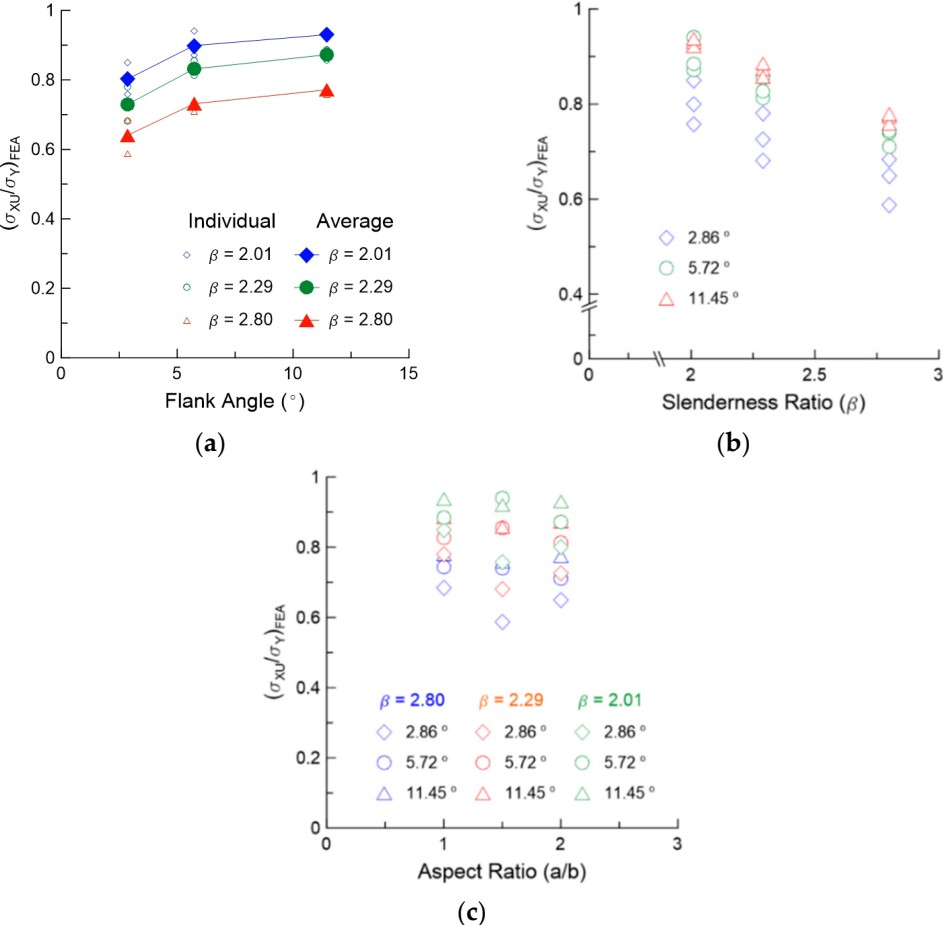

**Figure 12.** Ultimate strength of the curved plates based on the results of FE analysis: (**a**) effect of flank angle, (**b**) slenderness ratio, and (**c**) aspect ratio.

From a comparison with FE analysis, it can be seen that the ultimate strength of a curved plate obtained from experiment is significantly low when the slenderness ratio is 2.01. The collapse behavior and ultimate strength of a thin plate under longitudinal compressive loading is reported in previous studies [13,14]. In those studies, the secondary buckling after the primary buckling caused a serious decrease in the ultimate strength, accompanied by a change in the buckling mode. This behavior is because of the change in the in-plane stress distribution caused by a large deflection, and it occurs because of the snap-back phenomenon, which happens under conditions such as a specific slenderness ratio, low flank angle, and a certain aspect ratio. In this respect, it can be predicted that the curved plates with a certain slenderness ratio, 2.01, experienced secondary buckling with the increase in compressive load.

In addition, a previous study addressed the relationship between the curvature and ultimate strength as the aspect ratio increases [8]. This study found that the curvature increases the ultimate strength of a stiffened curved plate, especially when the plate has an intermediate aspect ratio (1.0 < a/b < 2.0). In this range, the ultimate strength of the stiffened curved plate increases rapidly at an aspect ratio of less than 1.0 and maintains a constant value at aspect ratio in excess of 2.0. Moreover, as the curvature increases, the aspect ratio has a greater effect on the ultimate strength, especially when the aspect ratio is less than 1.0, even though it is generally known that the aspect ratio has little influence on the ultimate strength of a curved plate under a longitudinal compressive load. Therefore, the ultimate strength of a curved plate with an aspect ratio of 1.5 can be regarded as being an unexpected phenomenon because this aspect ratio is the transition point between an

aspect ratio of 1.0 and 2.0, which determines the behavior characteristics of the ultimate strength of a curved plate.

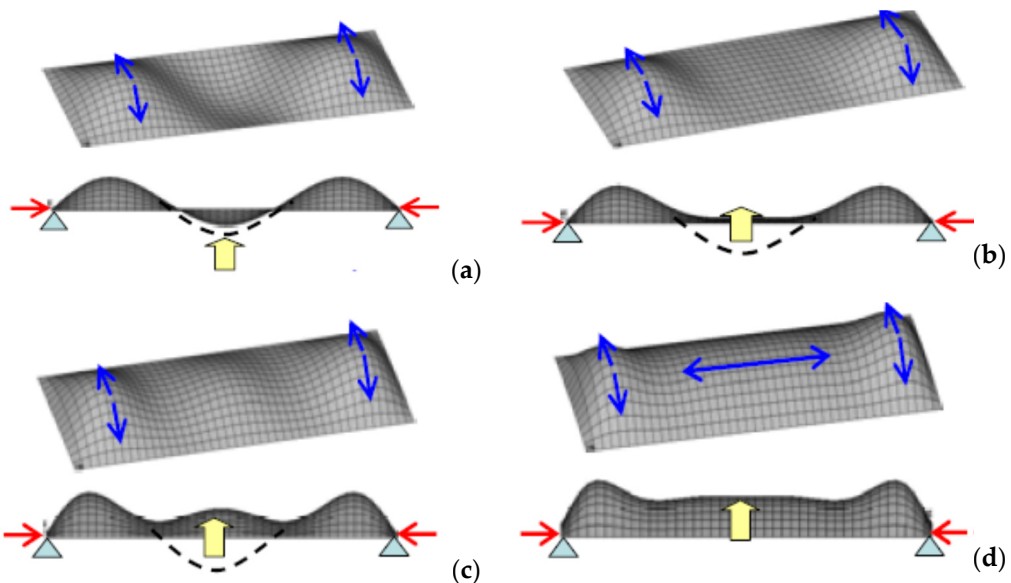

**Figure 13.** Elastic buckling mode under varying flank angles-(**a**) 2°, (**b**) 3°, (**c**) 5°, (**d**) 10° [1].

The initial imperfection might be another possible reason why the ultimate strength predicted by the FE analysis was quite different from that in the experimental results. In the experimental preparation process, the initial imperfection of the curved plate had not been measured. However, initial imperfection would have been implied, and to take this into account in the analysis stage, the Smith equation was applied to the FE model. As a result, a different initial imperfection was considered between the experiments and FE analysis. This could be one possible reason why the FE analysis showed a different trend in comparison to the experimental results. In the experimental preparation stages, the V-type groove, which was used to enable plate edge rotation, was designed to implement the simply supported condition. However, owing to the friction between jigs and plates, it is difficult to perfectly implement the moment free condition.

For these reasons, the values of the ultimate strengths obtained from the FE analysis and experiments are different. Figure 14 shows the comparison of ultimate strengths obtained from FE analysis and experiments for various values of flank angle and slenderness ratio. In general, for a curved plate under a longitudinal compressive load, the ultimate strength obtained from the FE analysis is higher than that obtained from experiments, while the ultimate strengths of flat plates are similar in both cases. The difference in ultimate strength in the two cases increases as the flank angle increases or as the slenderness ratio decreases (the plate thickness increase in case of present study). In other words, in the case of a thick curved plate with a large flank angle, the ultimate strength obtained from the FE analysis has low accuracy. In addition, structural design based on the FE analysis using the existing computational method can overestimate the real load capacity of a structure and cause the danger of an accident, such as the failure of a panel, because proper safety factor is not considered. Therefore, the empirical formula derived from the FE analysis should be modified based on the results of buckling tests. Based on these results, a new empirical formula for a curved plate under a longitudinal compressive load is suggested in the following section. The results derived by FE analysis and experiment are summarized in Table 4.

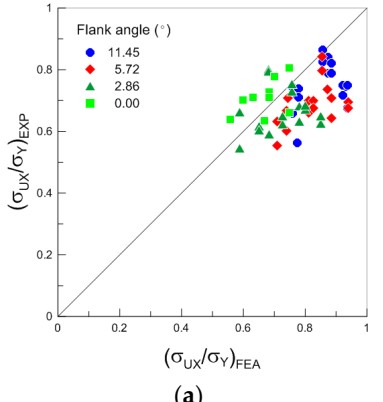
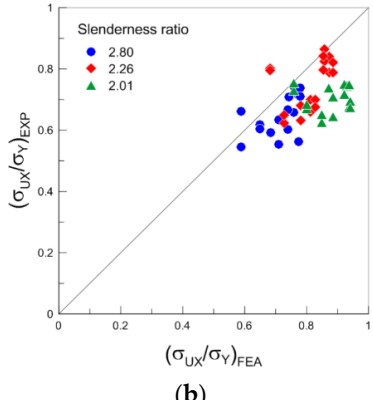

| (a) | (b) |

**Figure 14.** Comparison of FE analysis and experimental results: (**a**) effect of flank angle and (**b**) slenderness ratio.

**Table 4.** Experimental and FE analysis results.

| Test Model | β | b/R (rad.) | Ultimate Strength (MPa) | | FEA/Exp. |
|---|---|---|---|---|---|
| | | | Experiment | FEA | |
| F01 | | 0.00 | 218.98 | 190.97 | 0.8720 |
| C01 | 2.80 | 0.20 | 218.65 | 265.84 | 1.2158 |
| C02 | | 0.10 | 203.46 | 243.53 | 1.1969 |
| C03 | | 0.05 | 209.63 | 222.90 | 1.0633 |
| F02 | | 0.00 | 249.63 | 234.26 | 0.9384 |
| C04 | 2.29 | 0.20 | 279.29 | 299.71 | 1.0731 |
| C05 | | 0.10 | 233.11 | 278.64 | 1.1953 |
| C06 | | 0.05 | 218.15 | 249.21 | 1.1424 |
| F03 | | 0.00 | 226.42 | 257.06 | 1.1352 |
| C07 | 2.01 | 0.20 | 255.18 | 319.30 | 1.2513 |
| C08 | | 0.10 | 206.13 | 298.95 | 1.4503 |
| C09 | | 0.05 | 232.34 | 274.48 | 1.1813 |
| F04 | | 0.00 | 240.68 | 205.32 | 0.8530 |
| C10 | 2.80 | 0.20 | 240.28 | 271.73 | 1.1309 |
| C11 | | 0.10 | 217.47 | 253.84 | 1.1673 |
| C12 | | 0.05 | 207.13 | 201.67 | 0.9736 |
| F05 | | 0.00 | 243.40 | 216.51 | 0.8895 |
| C13 | 2.29 | 0.20 | 289.76 | 294.15 | 1.0151 |
| C14 | | 0.10 | 281.00 | 293.15 | 1.0433 |
| C15 | | 0.05 | 273.79 | 233.82 | 0.8540 |
| F06 | | 0.00 | 266.54 | 240.20 | 0.9011 |
| C16 | 2.01 | 0.20 | 251.66 | 315.99 | 1.2556 |
| C17 | | 0.10 | 234.66 | 322.47 | 1.3742 |
| C18 | | 0.05 | 254.34 | 259.92 | 1.0219 |
| F07 | | 0.00 | 217.27 | 228.90 | 1.0535 |
| C19 | 2.80 | 0.20 | 248.25 | 267.43 | 1.0772 |
| C20 | | 0.10 | 242.86 | 254.98 | 1.0499 |
| C21 | | 0.05 | 202.86 | 234.73 | 1.1571 |
| F08 | | 0.00 | 243.87 | 234.26 | 0.9606 |
| C22 | 2.29 | 0.20 | 275.95 | 303.82 | 1.1010 |
| C23 | | 0.10 | 235.86 | 283.76 | 1.2031 |
| C24 | | 0.05 | 225.30 | 267.77 | 1.1885 |
| F09 | | 0.00 | 276.15 | 257.06 | 0.9308 |
| C25 | 2.01 | 0.20 | 244.03 | 321.62 | 1.3180 |
| C26 | | 0.10 | 231.83 | 303.46 | 1.3090 |
| C27 | | 0.05 | 218.75 | 291.36 | 1.3319 |

*4.3. Suggestion for Design Formula*

To derive an empirical design formula, the expression of Frankland's formula with the revised coefficient with the same general form as the Faulkner and Guedes Soares expressions was used [15,16]. Based on the above results, the ultimate strength of a curved plate under longitudinal compression was empirically derived by curve fitting via the experimental results as follows.

$$\frac{\sigma_{UX}}{\sigma_Y} = \left(\frac{2.25}{\beta} - \frac{1.25}{\beta^2}\right) \times C_f \qquad \begin{array}{l} for\ 2.0 \le \beta \le 2.8 \\ for\ 3.0 \le \theta \le 11 \end{array} \tag{2}$$

$$C_f = \frac{C_a}{\beta^2} + \frac{C_b}{\beta} + C_c \tag{3}$$

$$\begin{aligned} C_a &= -299.06\left(\tfrac{b}{R}\right)^2 - 46.45\tfrac{b}{R} - 15.65 \\ C_b &= -272.81\left(\tfrac{b}{R}\right)^2 + 32.30\tfrac{b}{R} + 13.62 \\ C_c &= -59.43\left(\tfrac{b}{R}\right)^2 - 4.77\tfrac{b}{R} - 2.03 \end{aligned} \tag{4}$$

Similar to the work of Faulkner and Guedes Soares, the formula only works at a specific range of slenderness ratios and curvatures. However, it can be said that the present design formula leads to a very meaningful outcome in the sense that the design formula was derived on the basis of the experimental investigation of the various parameters of the curved plate, which is widely used in the ship and offshore industries. Figure 15 shows the variation in the ultimate strength plotted against the results predicted using the empirical formulae shown in Equations (2)–(4), as well as the experimental results. In the present formula, a correlation ratio of 0.5174 and a standard deviation of 0.3790 were obtained against the experimental results. The proposed empirical formula can be a good indicator for estimating the behavior of the curved plate.

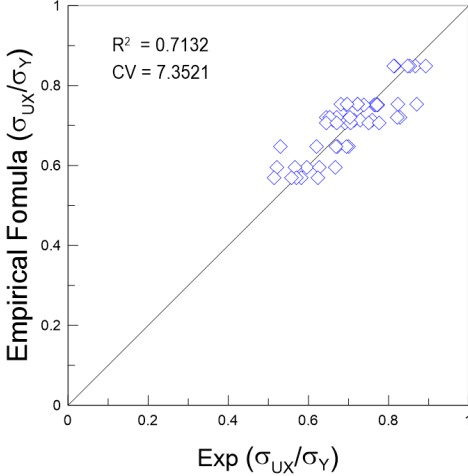

**Figure 15.** Correlation between the empirical formula with the ultimate strength of curved plates.

## 5. Conclusions

The objective of the present study was to investigate the ultimate strength and collapse behavior of a simply supported curved plate under longitudinal compressive load as well as complement the numerical results of the previous study. For this purpose, a series of buckling collapse experiment was carried out by considering flank angle, slenderness ratio, and aspect ratio of curved plates. In order to verify the effectiveness of the experimental results, an elastoplastic large deflection analysis was performed. Finally, new empirical design formulae were derived based on the experimental results for predicting the ultimate strength of curved plates. The inferences from the present study are summarized below.

- The ultimate strength of a simply supported curved plate increases with the increase in the flank angle and thickness. In addition, the ultimate strength of a curved plate is significantly low when the slenderness ratio is 2.01 because of special phenomenon such as secondary buckling.
- As the flank angle increases, an additional membrane stress, which is generated in the longitudinal direction at the central part of the curved plate, can restrain the lateral deflection and improve the strength of the curved plate.
- As the flank angle increases, the buckling mode of a simply supported curved plate sequentially changes in the order of two half-wave, three half-wave, and one half-wave under longitudinal compressive loads when aspect ratio is 2. The buckling mode are similar to those of the flat plates as the aspect ratio decreases.
- The difference in the ultimate strengths obtained from the FE analysis and experiment increased with the increase in flank angle, as well as with the decrease in slenderness ratio.
- Improved empirical formulae were derived based on the quantitative experimental database for predicting the ultimate strength of a curved plate. In addition, the result of correlation shows good agreement between the experimental results and the new design formulae. The results can be helpful as a reference in the design of curved plates.

**Author Contributions:** Conceptualization and project administration, J.-M.L.; test and writing, J.-H.K. and D.-H.P.; experimental review and editing, S.-K.K.; numerical simulation and editing, M.-S.K. All authors have read and agreed to the published version of the manuscript.

**Funding:** This research was a part of the project titled 'Development of the safety standards for marine hydrogen storage vessels and fuel supply systems', funded by the Ministry of Oceans and Fisheries, Korea. This work was supported by the R&D Platform Establishment of Eco-Friendly Hydrogen Propulsion Ship Program (No. 20006644) funded by the Ministry of Trade, Industry & Energy (MOTIE, Korea).

**Institutional Review Board Statement:** Not applicable.

**Informed Consent Statement:** Not applicable.

**Data Availability Statement:** The data presented in this study are available on request from corresponding author.

**Conflicts of Interest:** The authors declare no conflict of interest.

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
