# Peer review of "Experimental Study and Development of Design Formula for Estimating the Ultimate Strength of Curved Plates"

_applsci, doi:10.3390/app11052379_

Round 1
Reviewer 1 Report
- the current study aims to find the ultimate strength and collapse behaviour of simply supported curved plate under compressive stresses. The authors conduct buckling tests to evaluate the effect of flank angle, slenderness and aspect ratios of the curved plates. The authors also carry out elastic-plastic analysis to validate the experimental data. The study also proposed an empirical model which can be used to predict the ultimate strength of the curved plates. The empirical model showed good agreement with measured experimental data.
- The abstract needs more work, there is no mention of what are the most important findings from the study, this must be checked and added to the abstract.
- Move figure 1 to below line 99 from the introduction (suggested)
- Line 129 condition or conditions?
- Line 129 what does the authors mean by being conservative, do you mean to reduce computation costs in the mode? Or something else!
- Line 132 there are five references here, which study are you refereeing to from those references, please avoid using bulk citations unless they are given full credit elsewhere in the manuscript
- Why the compressive load for the 11.45 angle decreases significantly after certain displacement?
- This means that reducing the flank angle (eliminating it) would actually service better than having one for higher loads/displacements?
- Line 150 there is excessive use of the sentence “in this study or in the present study” please avoid using this as possible and just say for example in Line 150 The ultimate strength characteristics were investigated… check for this issue everywhere else in the manuscript
- Line 172 what do you mean by significant? Or you mean higher? If yes then use word higher not significant. And also explain why this happens and support your answer with references
- Figure 11 the authors need to plot the FE vs Exp in one graph to show us better the accuracy of the model?
- The authors also need to explain the reasons for the discrepancy between the FE and Exp results.
- Ok I see you did that in following pages…
- Abstract can be improved and discuss more about the FE model analysis.
Author Response
We greatly appreciate the reviewer’s valuable comments on our manuscript. We have responded to the various comments. Please see the attachment.

Reviewer 2 Report
Summary
The paper presents experimental and numerical studies on the compressive strength of cylindrical shells, compares the results and proposes an empirical formula to predict the ultimate compressive strength of such structural elements being part of curved panels of shops and offshore structures.
Broad comments
The paper presents valuable information for the study of curved plates under axial compression. The experiments cover a large range of the governing parameters and the description of the setup is adequate. Out-of-circularity imperfections are not presented which has implications on the comparison with FE analysis.
The proposal of empirical formula based on experiments is forced due to 2 reasons:
- The correlation coefficient is too low and the standard deviation is too high.
- Cf is too long and too precise (5 significant digits) to achieve such correlation and standard deviation. Does not make much scientific sense.
For these reasons the sentence ‘Therefore, the new empirical design formula presents a good agreement with the experimental results on curved plates.’ should be reformulated to reflect the reality.
Specific Comments
- 31: What do you mean by ‘continuous shipbuilding’? Eventually, remove ‘continuous’ or choose other words.
- 35 and others: ‘… Kim et al. investigate … are suggested [1].’ The option of authors was to put the reference at the end of the sentence far away from the name of the authors and it is consistent along with the text, which is good. A different criterion is normally used in literature: ‘… Kim et al. [1] investigate … are suggested.’ The authors may change the text accordingly but may maintain as it is if they prefer. If so, in this particular case and others, the reference is too far away from the name and should be moved to the end of the sentence in line 38.
- 52, 53. The statement is rather strong (serious accidents). The reports of accidents do not present any evidence of being originated by the failure of curved shells. Reformulate.
- 58: ulti/mate is ultimate.
- 132: ‘study’ is ‘studies’.
- Section 2.2: It is not properly described how it is obtained the simply supported condition on the lateral edges of the plate. Is it a similar setup as presented on the right side of fig. 3a).
- 165: ‘strengths’ is ‘strength’
- 173, Fig. 7: In this figure, one may note a lack of straightness on the lateral edges. These deformations on the edges occur after removal from jigs by relief of internal stresses or did they occur during the loading? Please comment in the text.
- 184-186: Eventually the shells have a different distribution of imperfections (out-of-circularity) along the surface and different dominant modes may result in big differences in strength. Since there is no information about local out-of-circularity is not possible to go any further. A comment on that would help to figure the reason for such behavior.
- 8b): moving the legend to the upper right corner benefits the reading of the figure.
- 201: ‘an equation’ is ‘equations’. The references of all CS that are used in the study should be mentioned.
- 233: 2 is upper position in the formula.
- Section 3.2: fig. 9 is not called in the text and there is no explanation or conclusion on it.
- 254: ‘… and a single half-wave is maintained when the flank angle is more than 11.45°.’ There is no information on the study that allows withdrawing such conclusion.
- 12 a): Individial is Individual
- 309-310: Eventually the main reason is that the shape of the initial imperfections on experiments are different than those used in FE analysis.
- Table 13: Units (MPa) are missing. One cannot understand what is in the last column; R is the curvature radii which is 0, 2000, etc. b/R should be theta so indicate (rad).
- 330: remove ‘new’. Frankland equation is older than the others. Please check.
Author Response

(The authors gave the same response as above.)

Reviewer 3 Report
This study deals with experimental and numerical studies to predict the ultimate strength of the curved plate with a lower flank angle. An excellent research outcome has been achieved by present a comparison with FE simulations. Authors could slightly revise the manuscript based on the following comments.
[Comment 1] Material properties: Authors could provide detailed material coupon test results.
[Comment 2] In the case of boundary condition shown in Fig 3(a), Can this actually be able to present the simply-supported boundary condition (B.C.)? Could this B.C. presents moment free condition? Perhaps, it might not be the ideal condition which means that the actual boundary condition might be in between simply-supported and clamped condition and it seems that this made the slight difference between test and FE results. It might be better to add some limitations to the test.
[Comment 3] Initial deflection of the plate: Smith et al. (1988) proposed three levels of initial deflection of the plate such as slight (0.025*beta^2*t), average (0.1*beta^2*t), and severe level (0.3*beta^2*t).
What does it stand for “normal level” presented in Line No. 233?
[Comment 4] FE (a material property): The author could add some figures or tables to explain the material curve used in FE simulations.
[Comment 5] It might be helpful if the obtained load carry capacity curves (from the test) in the Appendix. From the current version, the selected one (1) case result only plotted in Fig 6(a), but it might not be challenging to prepare another 7-8 graphs based on Table 1.
[Comment 6] Accuracy of the obtained formula: Authors could add R^2 and COV values in Fig. 15 so that readers could understand the accuracy of the proposed formula.
[Comment 7] If authors measure the amounts of initial deflection, it could also be provided in the table.
Besides, please check the number of tests performed. They mentioned that 72 cases had been conducted, but only 32 cases are shown in the manuscript.
Author Response

(The authors gave the same response as above.)

Round 2
Reviewer 1 Report
Reviewer’s comment (4): still not answered!
all other questions were answered
Author Response
We greatly appreciate the reviewer’s valuable comments on our manuscript. Please see the attachment.
